# Bioengineering of *Escherichia coli Nissle 1917* for Production and Excretion of Spermidine, a Key Metabolite in Human Health

**DOI:** 10.3390/metabo12111061

**Published:** 2022-11-02

**Authors:** Clément Caffaratti, Caroline Plazy, Valérie Cunin, Bertrand Toussaint, Audrey Le Gouellec

**Affiliations:** 1University Grenoble Alpes, CNRS, UMR 5525, VetAgro Sup, Grenoble INP, CHU Grenoble Alpes, TIMC, 38000 Grenoble, France; 2Plateforme de Métabolomique GEMELI-GExiM, Service de Biochimie, Biologie Moléculaire et Toxicologie Environnementale, UM Biochimie des Enzymes et des Protéines, Institut de Biologie et Pathologie, CHU Grenoble-Alpes, 38000 Grenoble, France

**Keywords:** microbiota, immunity, spermidine, metabolic engineering, probiotics, Live Biotherapeutic Product, metabolomics

## Abstract

Microbiota-derived metabolites have biological importance for their host. Spermidine, a metabolite described for its protective effect in age-related diseases, is now studied for its role in the resolution of inflammation and gut homeostasis. Strategies to modulate its production in the gastrointestinal tract are of interest to increase host spermidine intakes. Here, we show that metabolic engineering can be used to increase spermidine production by the probiotic *Escherichia coli Nissle* 1917 (EcN), used in humans. First, we found that increasing the expression of genes involved in polyamine biosynthesis, namely the S-adenosylmethionine synthase *speD* and the spermidine synthase *speE*, resulted in an increase in spermidine produced and excreted by our engineered bacteria. The major drawback of this first attempt was the production of acetylated forms of spermidine. Next, we propose to solve this problem by increasing the expression of the spermidine exporter system MdtI/MdtJ. This second strategy had a major impact on the spermidine profile found in the culture supernatant. Our results demonstrate, for the first time, the feasibility of rationally engineering bacterial probiotic strains to increase their ability to deliver the microbiota-derived metabolite, spermidine. This work illustrates how metabolomic and synthetic biology can be used to design and improve engineered Live Biotherapeutic Products that have the potential to be used in personalized medicine.

## 1. Introduction

The human intestine is colonized by trillions of microorganisms which participate in the regulation of the host’s physiology. There is increasing evidence of microbiota’s involvement in the regulation of human physiology and in the development of human pathologies, such as diseases involving inflammatory processes. In the gastrointestinal tract, the microbiota are involved in a constant dialogue with the host immune system. Notably, metabolites derived from the microbiota are essential to this dialogue, as they are involved in immune regulation [1]. Their role in the maintenance of the intestinal barrier is now well described, showing the importance of preserving the microbiota and its functionality. An imbalance within this ecosystem, described as dysbiosis, leads to a loss of microbial diversity and is accompanied by a depletion of major metabolites that play a role in maintaining intestinal and immune homeostasis.

Manipulation of the intestinal microbiome is proposed as a potential strategy to prevent or treat a wide range of human diseases [2]. Current strategies rely primarily on the use of probiotics, prebiotics, fecal microbiota transplantation, symbiotic microbial consortia or microbiota-derived proteins and metabolites. Some probiotic strains are now examined for specific pathologies and are referred to as Live Biotherapeutic Product (LBP). LBPs are defined as “*a biological product that (1) contains live organisms, such as bacteria; (2) is applicable to the prevention, treatment, or cure of a disease or condition of human beings; and (3) is not a vaccine*” by the Food and Drug Administration (FDA).

Some industries or academic laboratories are also developing a new class of LBP: engineered LBPs. Among them, engineered LBPs encompass engineered bacterial therapeutics that have been engineered by adding or deleting genetic material within the organism. They are designed using synthetic biology to propose a solution for unmet medical needs. For instance, engineered LBPs can be rationally designed to restore a missing biological function [3] or be used as cargo to deliver biological medical products [4]. In a recent paper, 13 engineered LBPs were listed to be in clinical development (Phase 1/2) [5].

*Escherichia coli* Nissle 1917 (EcN) is a probiotic isolated in 1917 by the German physician Alfred Nissle and is currently commercialized under the product name Mutaflor^®^ by AdeyPharm. The anti-inflammatory properties of EcN have been evaluated in a substantial number of mice colitis models [6,7,8]. This is attributable to EcN’s ability to reduce intestinal permeability and maintain intestinal barrier function by up-regulating tight junctions in intestinal epithelial cells (IEC) and increasing colonic regulatory T cells (Treg). Furthermore, in humans, EcN has been recommended to treat inflammatory bowel disease and irritable bowel syndrome. For instance, it has been recognized by the European Crohn’s and colitis organization to be as effective as mesalazine in maintaining remission of ulcerative colitis [9,10], and the ESPEN recommendations inform clinicians that probiotic therapy should be considered for the maintenance of remission in UC [11].

Polyamines (PAs) are organic compounds having more than two amino groups. Spermidine (N3) and spermine (N4) are two putrescine derivatives found in human cells. In the gut, it is accepted that PAs reaching the small intestine come from the diet and are rapidly absorbed. In contrast, PAs in the lower intestine are derived from the metabolic activity of the gut microbiota [12,13]. Previous research has shown that arginine supplementation, a polyamine precursor, affects PAs synthesis derived from the gut microbiota [14], demonstrating the importance of the microbiota on the quantity of PAs in the colonic lumen.

More recently, multiple beneficial effects of spermidine have been reported on epithelial and immune function. Spermidine maintains the barrier function by having a pro-proliferative role on the colonocytes, which promotes its renewal [15]. Moreover, spermidine is described as an anti-inflammatory metabolite of both innate and adaptive immunity. Spermidine has been reported to be a negative regulator of IFNγ-induced responses through the up-regulation of the protein tyrosine kinase non-receptor type 2 (PTPN2) in monocytes [16]. As well, spermidine was shown to play a major role in regulating T-cell differentiation and function. Spermidine exposition potentiates the in vitro Foxp3^+^ T-cell differentiation from both naïve and Th17 CD4^+^T-cells in an autophagy-dependent manner [17]. Therefore, it seems important to maintain a stable spermidine pool in the gut to support homeostasis and gut health.

In this work, using synthetic biology applied to the well-characterized probiotic EcN, we rationally designed a metabolically engineered strain to increase the amount of spermidine produced. We characterized the effect of this engineering on the growth of EcN and on its global metabolism by untargeted metabolomics. Finally, we characterized the impact of our engineering on spermidine production.

## 2. Materials and Methods

### 2.1. Bacterial Strains and Growth Conditions

Probiotic *E. coli* Nissle 1917 (EcN) was obtained from Ardeypharm. The growth was performed in the lysogeny broth Miller (LB-Miller) medium (Cat #0103-AthenaES™, USA). Briefly, a seed culture was prepared for each condition by inoculating a single CFU from a fresh plate into 3 mL LB medium and cultured overnight at 37 °C. The seed culture was used (100 µL) to inoculate a 100 mL Erlenmeyer flask containing 10 mL of LB medium. Then, the Erlenmeyer flasks were placed at 37 °C and 250 rpm in a shaking incubator (Thermo Scientific™ MaxQ™ 6000, Waltham, MA, USA). Cell growth was monitored by measuring the absorbance at 600 nm (OD600) with an Eppendorf BioSpectrometer^®^ spectrophotometer.

### 2.2. Plasmids Construction

All plasmids in this study were constructed using the pUCBB-ntH6-eGFP (Addgene #32557, USA), described in the paper by Vick et al. [18]. The multiple cloning site (MCS) was used to change several elements of the biological circuit, as follows: Promoter + Ribosome binding site (*Eco*RI–*Nde*I); Gene: *Nde*I–*Nsi*I; Terminator (if added): *Nsi*I-*Spe*I.

The *speD* and *speE* genes were amplified by standard PCR on the *E. coli* W3110 genome and assembled into the pUCBB-ntH6-eGFP plasmid with a restriction digest/Gibson cloning strategy (HiFi DNA Assembly Master Mix, New England Biolabs). Briefly, an initial PCR was performed with the primers RBS-*speE*-F and RBS-*speE*-R to add an optimized ribosome-binding site (RBS) sequence to the 5′ region of the *speE* gene. RBS sequences were obtained using the UTR Designer Tool [19]. Then, *speD* and *speE* were finally amplified with primers designed to create the overlapping sequence required for Gibson assembly. The *mdtI* and *mdtJ* genes were amplified by standard PCR on the *E. coli* W3110 genome. Primers used in this study are available in Table 1.

### 2.3. Sample Collection and Sample Preparation

During the culture, 400 µL was collected at the beginning and 24 h after the start of the culture. Each bacterial suspension was then centrifuged at 10,000× *g* for 10 min at 4 °C. Afterwards, the supernatant was collected and stored at −80 °C while waiting to follow the sample preparation protocol.

For sample preparation, we used a protein precipitation strategy previously described [20]. Briefly, each supernatant was treated with methanol (80% methanol, 20% Sample) and mixed using a vortex mixer for 20 s, left on ice at 4 °C for 30 min to allow protein precipitation, then centrifuged for 20 min at 13,000× *g*. After centrifugation, the supernatant was collected and subjected to desiccation before being stored at −80 °C for analysis.

### 2.4. Liquid Chromatography Coupled with High-Resolution Mass Spectrometry (LC-HRMS) Analysis

Untargeted metabolomic profiling of the culture supernatant was conducted using ultra-high-performance liquid chromatography (Vanquish Flex, Thermo Fisher Scientific, Waltham, MA, USA) coupled with a Q Exactive Plus Orbitrap mass spectrometer (Thermo Fisher Scientific, Waltham, MA, USA).

The chromatographic separation was carried out on a C18 Accucore™ RP MS (2.1 mm × 150 mm, 2.6 μm, Thermo Fisher Scientific, Waltham, MA, USA) at 30 °C with a flow elution rate of 400 μL/min.

For the acquisition presented in the result part 3.1, we used the following protocol:The mobile phases consisted of A (100% water + 0.1% formic acid) and B (100% acetonitrile (ACN) + 0.1% formic acid). Elution started with an isocratic step of 3 min at 1% mobile phase B, followed by a linear gradient from 1% to 100% mobile phase B for the next 4 min. These proportions were kept constant for the next 3 min before returning to 1% B for 4 min.

For the acquisition presented in the result part 3.2, we used the following protocol:The mobile phases consisted of A (100% water + 0.1% formic acid) and B (100% acetonitrile (ACN) + 0.1% formic acid). Elution started with an isocratic step of 2 min at 1% mobile phase B, followed by a linear gradient from 1% to 100% mobile phase B for the next 12 min. These proportions were kept constant for the next 4 min before returning to 1% B for 5 min.

The mass spectrometer was fitted with an electrospray source (ESI) operating in positive modes. It was operated with a capillary voltage at 5 kV in the positive ionization and a capillary temperature set at 280 °C. The temperature of the autosampler compartment was set at 4 °C, and the injection volume was 5 μL. Detection was carried out from m/z 60 to 900. During the acquisition, we used a quality-control (QC) pooled samples strategy to ensure that measurements and peak detection were stable. This QC sample was injected every 10 injections. Features with a coefficient of variation (CV) higher than 30% were removed from the analysis.

### 2.5. LC-HRMS Data Processing

Raw LC-HRMS data were converted to m/z extensible markup language (.mzXML) in centroid mode using MSConvert ProteoWizard (release version 3.0.20350-23fa10407). Preprocessing of the data was performed on MzMine 2.53 [21]. Normalization and statistical analyses were performed on MetaboAnalyst 5.0 [22]. Mass detection was realized keeping the noise level at 5.0 × 10^6^ for MS level 1 and 1.0 × 10^6^ for MS level 2. An ADAP chromatogram builder was employed, using a minimum group size of scans of 4, a group intensity threshold of 3.0 × 10^6^, and an m/z tolerance of 0.001. An ADAP wavelet deconvolution algorithm was used with the following standard settings: S/N threshold of 10, coefficient/area threshold of 20, peak duration range of 0.00–4 min and RT wavelet range of 0.01–0.10. MS2 scans were paired using an m/z tolerance range of 0.01 Da and a RT tolerance range of 0.2 min. Isotopes were grouped using the isotopic peak grouper algorithm (m/z tolerance of 5 ppm, RT tolerance of 0.2 min, maximum charge of 3). For the peak alignment, this step was performed using the Join aligner module (m/z tolerance of 5 ppm and weight of 75 and 25 were attributed for m/z and RT, respectively; RT tolerance of 0.8 min).

### 2.6. Metabolite Annotation

Metabolites identification of putrescine and spermidine was performed by injecting authentic chemical standards (Sigma, St Louis, MO, USA) under the same analytical conditions. In the absence of an available authentic chemical standard, metabolites of interest were only considered as putatively annotated based on accurately measured masses and interpretation of the MS/MS spectra when available, as described by Aros-Calt et al., 2015 [23]. First, feature annotation was performed by using fragmentation spectra in the spectral library of the Global Natural Product Social Molecular Networking (GNPS) platform [24]. The Library Search feature of GNPS allowed us to identify N8-acetylspermidine (GNPS Library Spectrum CCMSLIB00006122363). In a second attempt to perform metabolite identification, SIRIUS and CSI: FingerID [25] were used. SIRIUS identifies metabolites based on their MS/MS spectra and by creating fragmentation trees that can explain the fragmentation spectrum. Then, CSI:FingerID uses the fragmentation tree to predict the compound’s molecular fingerprint. Using this second approach, we obtained a similarity score of 100 for N8-acetylspermidine and 92.1 for diacetylspermidine (see Appendix A).

### 2.7. Statistical Analysis

Results in figures are represented as mean per group ± standard deviation (SD). The statistical test used for each hypothesis is described in the figure legend. The individual data were obtained from eight independent cultures. All statistical analyses were performed on GraphPad Prism ver.9.0 for Windows (GraphPad Software, San Diego, CA, USA) and MetaboAnalyst 5.0. Results were considered significant as follows: * *p* < 0.05.

## 3. Results

### 3.1. Overexpression of the Genes speD and speE Increases the Amount of Spermidine

In *E. coli*, the flux of PAs depends on import, export, synthesis and catabolism (Figure 1A). Putrescine synthesis relies on ornithine or arginine decarboxylation performed, respectively, by ornithine decarboxylase (SpeC) or arginine decarboxylase (SpeA) and agmatinase (SpeB) [26,27]. Spermidine synthesis begins with the synthesis of decarboxylated S-adenosylmethionine, which depends on the conversion of methionine to S-adenosylmethionine (SAM) and its decarboxylation by the enzyme SAM decarboxylase (SpeD) [28]. Once putrescine is produced, it is used to produce spermidine with the spermidine synthase SpeE, where decarboxylated SAM serves as a substrate in this reaction [29].

To increase the synthesis of Spermidine in EcN, we first decided to increase the amount of the enzymes SpeD and SpeE that catalyze the last steps of the synthesis of spermidine. For that purpose, the genes *speD* and *speE* were assembled into the plasmid pUCBB-ntH6-eGFP, an optimized BioBrick™ vector, derived from the pUC backbone [18]. We constructed pCC25, a plasmid where *speD* and *speE* were placed under the control of the P*_lacP’_* promoter, and pCC26, a plasmid where *speD* and *speE* transcription was regulated by the P*_fnrs_* promoter and where an optimized RBS was used. P*_lacP’_* is a constitutive promoter derived from the lac promoter, and P*_fnrs_* is an anaerobic inducible promoter regulated by the fumarate and nitrate reductase (FNR) transcription factor [30]. The P*_fnrs_* promoter was chosen because it has already shown its interest in anaerobic environments, such as the gut [3]. After designing these plasmids, we cultured the EcN, EcN pCC25 and EcN pCC26 strains in an LB medium and monitored their growth. We did not see any difference in the growth curves between the three groups (Figure 2A). We then used RT-qPCR to relatively quantify the gene expression of *speD* and *speE*, and we observed an overexpression of *speE* and *speD* genes (Appendix A). Then, to look further for the effect of these overexpressions, we collected culture supernatants at times 0 and 24 h from 8 independent cultures and analyzed them by LC-MS/MS (see the Methods section and reference papers [12,31]). As shown in Figure 2B, EcN, transformed with the plasmid pCC25 and pCC26, have increased spermidine concentrations in their supernatants. In our experiment, the putrescine concentration increased in all conditions between 0 h and 24 h. However, the amount of putrescine present at the end of the cultures was lower if the strains were transformed either with pCC25 or pCC26 (Figure 2C). The putrescine and spermidine concentration profiles between wild type EcN and engineered EcN suggests that EcN are releasing putrescine into the extracellular medium. The overproduction of SpeD/E tends to limit the amount of putrescine present at the end of the culture, indicating that there is a higher consumption of putrescine. However, the rise in spermidine was also accompanied by a significant increase in acetylated forms of spermidine, as shown by the study of the mono-acetylated and diacetylated forms (Figure 2D,E).

### 3.2. Putrescine Supplementation of the Medium Increases Spermidine Production

With our previous experiment, we identified that putrescine is more consumed by our engineered EcN strains. We, therefore, asked whether the substrate, putrescine, was limiting for spermidine biosynthesis, and we then cultured EcN, EcN pCC25 and EcN pCC26 with LB media supplemented with 1 mM putrescine. This concentration is a typical concentration of putrescine in the intestinal lumen of healthy humans [32]. As shown in Figure 3B,C, we reconfirmed our observations, showing that EcNs transformed with plasmid pCC25 or pCC26 have increased concentrations of spermidine and decreased concentration of putrescine in their supernatants. Interestingly, putrescine supplementation results in the increase in spermidine compared with the unsupplemented medium in EcN pCC25 and EcN pCC26 (Figure 3D). However, the addition of putrescine did not result in an increase in the production of spermidine by the wild type EcN. In conclusion, the metabolic engineering of the strain (EcN pCC25 and EcN pCC26) combined with the supplementation of LB medium maximizes the amount of spermidine produced.

### 3.3. Excretion and Production of Acetylated Forms of Spermidine Can Be Limited by Overexpressing the Spermidine Exporter MdtI/MdtJ

We demonstrated that increasing the concentration of SpeD and SpeE is an effective strategy to increase the amount of spermidine available in the extracellular compartment. However, as we have shown previously, this accompanies an increase in the production and excretion of acetylated forms of spermidine. This increase makes sense considering the biology of PAs in *E. coli* and the mechanisms by which this species regulates intracellular polyamine levels. As it was described before, spermidine accumulation is toxic for *E. coli* [33]. Two strategies are used by these bacteria to limit spermidine toxicity, namely, spermidine overexcretion using specific transporters [34] or spermidine acetylation using the spermidine N(1)-acetyltransferase SpeG. Nevertheless, since our goal is to maximize the amount of spermidine in the extracellular compartment, it may be interesting to promote spermidine excretion to limit the amount of acetylated spermidine. To address this issue, we have added a second biological circuit on pCC25 and pCC26 to increase the production of the two-component exporter system MdtI/MdtJ. This new circuit was composed of the constitutive promoter P*_J23102_*, an optimized RBS and the coding sequence of MdtI and MdtJ. The two new plasmids were designed and assembled and named plasmid pCC31 (P*_fnrs_*-*speD*-*speE*-B0015-P*_J23102_*-*mdtJ*-*mdtI*) and plasmid pCC35 (P*_LacP’_*-*speD*-*speE*-B0015-P*_J23102_*-*mdtJ*-*mdtI*). Each plasmid was transformed into EcN, and we used the LB media supplemented with putrescine and the same methodology to assess whether the addition of the MdtIJ transporters improved spermidine production and excretion. Metabolic profiling of the different strains showed that the addition of the MdtI/MdtJ biological system induced a major change in the metabolite levels of the PAs pathway. Indeed, spermidine is significantly increased in EcN pCC31 and EcN pCC35 compared with EcN (fold change: 3 and 5, respectively) or EcN transformed with pCC25 or pCC26 (fold change: 2.8 between EcN pCC35 and EcN pCC25) (Figure 4D). Furthermore, the accumulation of acetylated forms of spermidine observed with plasmids pCC25 and pCC26 disappeared in strains transformed with plasmids pCC31 and pCC35 (Figure 4E,F). These observations suggest that these engineering strategies profoundly impacted the metabolites fluxes belonging to the PAs family, in favor of spermidine production and excretion.

### 3.4. EcN Engineering Seems to Have a Limited Impact on Its Overall Metabolism

We used an untargeted LC-MS/MS metabolomics approach to pinpoint the main differences in the metabolome of the engineered EcN and the wild type. In all, 178 features were detected and met our quality requirements presented in the method section, and volcano plot analysis was used to compare EcN pCC25 and EcN pCC35 with EcN (Figure 5). Analysis of the EcN pCC25 and pCC35 strains showed that the majority of features were unchanged at 24 h if the strains’ supernatants were respectively compared to the EcN strain (Figure 5A,B). This first observation highlights the fact that this metabolic engineering did not affect the global metabolism of the EcN pCC25 and pCC35. However if we look in detail at the effect of the addition of pCC25 and pCC35 to the wild type strain, we can see that the abundance of five features was decreased (putrescine, methionine and three features related to N-Acetyl-isoleucine) and seven increased (spermidine, diacetyl-spermidine, acetyl-spermidine and four features unidentified) in EcN pCC25 supernatant compared to EcN (Figure 5A), while in pCC35, the abundance of eight features (arginine, acetyl-putrescine, methionine, alanyl-valine and, four features were unidentified) was decreased and seven increased (spermidine, N-Acetyl-spermidine, and five features were unidentified) in its supernatant compared with EcN (Figure 5B). In conclusion, the main metabolic changes observed by this unsupervised method demonstrate that the changes brought about by both types of engineering (pCC25 and pCC35) are related principally to the polyamines pathway.

## 4. Discussion

The gut microbiota produce a wide variety of metabolites that can be absorbed directly from the colon or reach systemic circulation. Some of these metabolites appear to play a key role in host health and disease. Accumulating evidence shows that bioactive molecules, such as PAs, act directly on host cells to perform physiological functions [35]. Research on spermidine has mainly focused on its anti-aging, neurologic and cardiovascular benefits, but convincing evidence has highlighted its impact on the epithelium of the intestinal mucosa and intestinal immunity [15,17]. In humans, spermidine intake is mainly dependent on diet and microbiota. Two strategies were then proposed: the first one consists of proposing supplementation in spermidine or a diet rich in spermidine while the other one proposes to modulate intakes of spermidine in the colon by modulating the gut microbiota. Both strategies have shown that increased spermidine intake results in an anti-inflammatory response and improved gut homeostasis.

We hypothesize that modifying the spermidine supply with genetically engineered bacterial strains is an attractive strategy. In this study, we used the probiotic EcN, recognized for its anti-inflammatory properties, to perform metabolic engineering that will increase its spermidine production. In a first attempt, we increased the expression of *speD* and *speE* and were able to show that this overexpression already resulted in a considerable increase in the amount of spermidine produced and transported to the extracellular compartment. One of the main problems encountered with this strategy was the accumulation of acetylated forms of spermidine, which was, therefore, an important issue to maximize the amount of spermidine produced and bioavailable in the exometabolome. Based on accumulated knowledge of Pas’ metabolic pathways in *E. coli*, we hypothesized that this accumulation was related to a normal response of the bacterium to limit PA toxicity. We have abolished this effect by increasing the expression of the two-component exporter system MdtI/MdtJ. This strategy significantly increased the amount of spermidine present in the extracellular space while limiting the acetylation of spermidine by EcN. We have also showed that spermidine production was influenced by the availability of putrescine in the environment of the bacteria. Finally, we evaluated the impact of our engineering on the metabolomes of the strains. The data generated showed that the metabolic engineering we performed has a limited impact on the overall metabolome of the strains and that this impact seems to target the metabolites related to the PA pathway. Although it provides valuable insights to answer these questions, additional experiments are needed to maximize the metabolic coverage being detected.

To our knowledge, this study provides the first evidence that spermidine can be overproduced in a probiotic bacterial strain. This strategy will allow researchers to address complex questions regarding the impact of targeted variations in a metabolic pathway on the anti-inflammatory effects of EcN and, more importantly, the impact of these engineered strains on the microbial ecosystem of the gut. As PA are bioactive molecules that are important in maintaining human health, this engineered probiotic will be useful to explore more deeply the importance of the PA on the resolution of pathological inflammatory processes, as is the case in chronic inflammatory bowel diseases.

## 5. Patents

We have filed a patent application with the registration number FR2209825.

## Figures and Tables

**Figure 1 metabolites-12-01061-f001:**
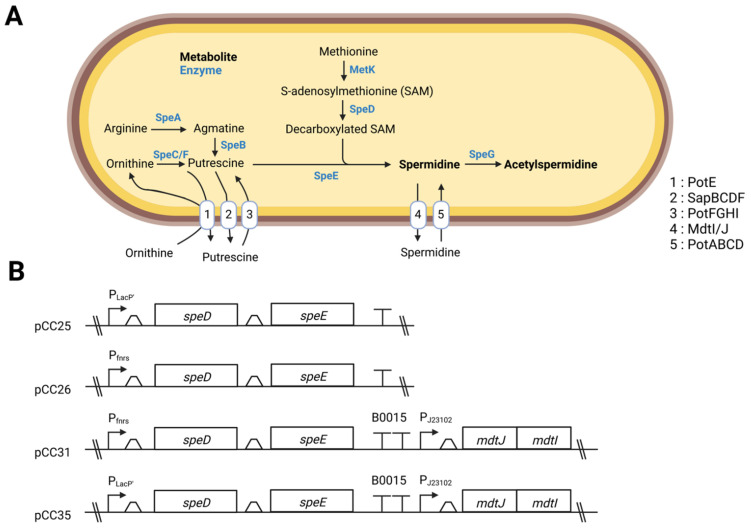
(**A**) Polyamine synthesis pathways in EcN. List of transporters and enzymes: MdtI/J, spermidine export proteins; PotABCD, ATP-binding spermidine-preferential uptake system; PotFGHI, ATP-binding putrescine-specific uptake system; PotE, putrescine/ornithine antiporter; SapABCDF, Putrescine export system permease; SpeA, biosynthetic arginine decarboxylase; SpeB, agmatinase; SpeC/SpeF, biosynthetic/degradative ornithine decarboxylase; SpeE, spermidine synthase; SpeG, spermidine acetyltransferase (**B**) General outline of constructions presented in this study. Created with BioRender.com.

**Figure 2 metabolites-12-01061-f002:**
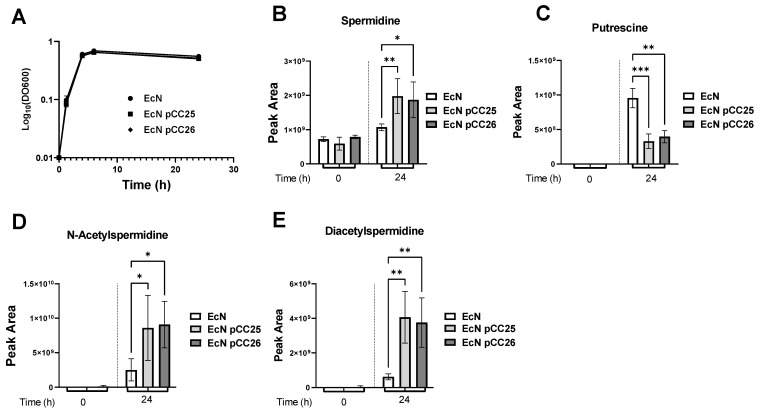
Production of spermidine by EcN pCC25 and EcN pCC26 compared to the wild type EcN. (**A**) Growth curves of EcN, EcN pCC25 and EcN pCC26 in LB medium (n = 8). (**B**–**E**) Peak area of spermidine (**B**), putrescine (**C**), N-acetylspermidine (**D**) or diacetylspermidine (**E**) measured by LC-MS in the culture supernatant of EcN, EcN pCC25 and EcN pCC26 at the beginning of the culture (0) or 24 h after the cultivation start (24) (n = 8). The individual data were obtained from independent cultures. All data shown represent the mean ± SD. Statistical significance was calculated using Kruskal–Wallis test and Benjamini, Krieger and Yekutieli Multiple comparisons. Statistical significance indicated as * *p* < 0.05, ** *p* < 0.01, *** *p* < 0.001.

**Figure 3 metabolites-12-01061-f003:**
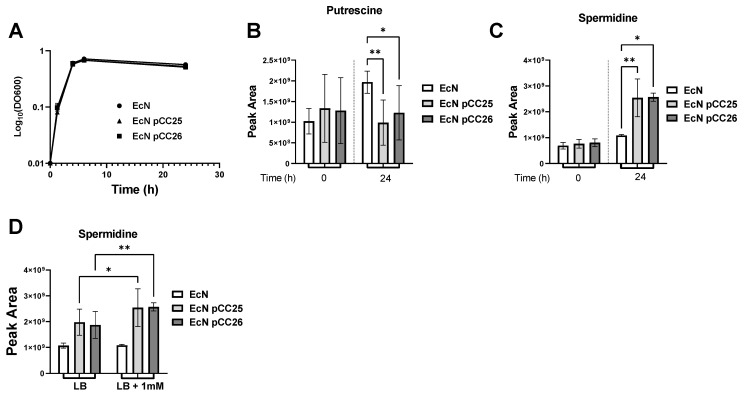
Targeted metabolomic analysis of EcN pCC25 and EcN pCC26 compared with EcN WT in LB broth supplemented with 1mM of putrescine. (**A**) Growth curves of EcN, EcN pCC25 and EcN pCC26 in LB medium supplemented with 1mM of putrescine (n = 8). (**B**,**C**) Peak area of spermidine (**B**) and putrescine (**C**) measured by LC-MS in the culture supernatant of EcN, EcN pCC25 and EcN pCC26 at the beginning of the culture (0) or 24 h after the cultivation start (24) (n = 8). (**D**) Comparison of the signal obtained for spermidine at 24 h between our two culture conditions (LB and LB supplemented with 1 mM putrescine). The individual data were obtained from 8 independent cultures. All data shown represent the mean ± SD. Statistical significance was calculated using Kruskal–Wallis test and Benjamini, Krieger and Yekutieli Multiple comparisons. Statistical significance indicated as * *p* < 0.05, ** *p* < 0.01.

**Figure 4 metabolites-12-01061-f004:**
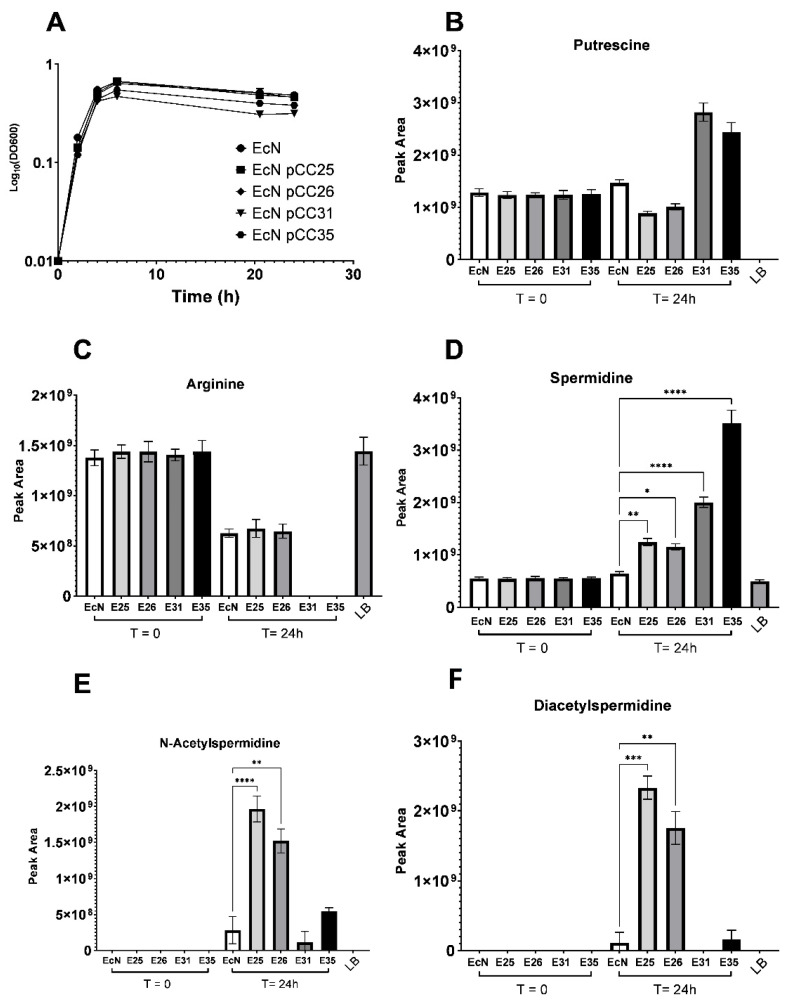
Targeted metabolomic analysis of EcN pCC25, pCC26, pCC31 and pCC35 compared with EcN WT in LB broth supplemented with 1mM of putrescine. (**A**) Growth curves of EcN, EcN pCC25, EcN pCC26, EcN pCC31 and EcN pCC35 in LB medium supplemented with 1mM of putrescine (n = 8). (**B**–**F**) Peak area of the metabolite involved in the metabolic pathways of spermidine (putrescine (**B**), arginine (**C**), spermidine (**D**), N8-acetylspermidine (**E**) and diacetylspermidine (**F**) used to estimate the amount of each metabolite in the culture supernatant of our strains at the beginning of the culture (0) or after 24 h (24) (n = 8). The individual data were obtained from independent cultures. All data shown represent the mean ± SD. Statistical significance was calculated using Kruskal–Wallis test and Benjamini, Krieger and Yekutieli Multiple comparisons. Statistical significance indicated as * *p* < 0.05, ** *p* < 0.01, *** *p* < 0.001, **** *p* < 0.0001.

**Figure 5 metabolites-12-01061-f005:**
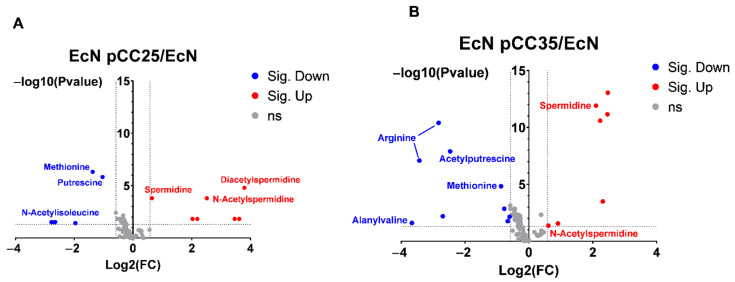
Volcano plot analysis of metabolic changes between EcN pCC35 and EcN pCC25 compared with EcN at 24 h of culture. (**A**) Volcano plot comparing EcN pCC25 versus EcN metabolic profile cultured in LB and collected at 24 h. (**B**) Volcano plot comparing EcN pCC35 versus EcN metabolic profile cultured in LB and collected at 24 hours. Each point on the volcano plots is based on the *p*-value and the fold change. Cutoff for the *p*-value is <0.05; fold change cutoff is >1.5 or <0.66. All *p*-values were adjusted using the False Discovery Rate method. All points were annotated on the graph when feature identification was possible using GNPS or SIRIUS.

**Table 1 metabolites-12-01061-t001:** Primers used in this study. *speE* (spermidine synthase) and *speD* (S-adenosylmethionine synthase). *mdtJ* and *mdtI* genes code for the spermidine export proteins MdtIJ.

Name	Primer Sequence (5′-3′)
P*_fnrs_*-F	5′ ATTGAATTCATCTAGAAAAAACGCCGCAAAGTT 3′
P*_fnrs_*-R	5′ ATTCATATGTTTATTCTTCCCTCCTCTATC 3′
*speD*-F	5′ TAATTTAAGGGGGTAACATAATGTTGAAAAAACTGAAACTGC 3′
*speD*-R	5′ TCCGTCCTTACGTCCCCACTTTAAACAGCGGGCATATTGC 3′
RBS-*speE*-F	5′ AGTGGGGACGTAAGGACGGATTTGGATGGCCGAAAAAAAACAGT
RBS-*speE*-R	5′ TTAGGACGGCTGTGAAGC 3′
*speE*-F	5′ GCAATATGCCCGCTGTTTAAAGTGGGGACGTAAGGACGGA 3′
*speE*-R	5′ TTATTTGATGCCTGGATGCATTAGGACGGCTGTGAAGCCA 3′
B0015-F	5′ ATTATGCATCCAGGCATCAAATAAAACG 3′
B0015-R	5′ ATTACTAGTTATAAACGCAGAAAGGCCC 3′
*mdtJ-mdtI*-F	5′ ATTCATATGTATATTTATTGGATTTTATTAGGTCTG 3′
*mdtJ-mdtI*-R	5′ ATTACTAGTTCAGGCAAGTTTCACCAT 3′
oSeq1	5′ CTGACGTCTAAGAAACCATT 3′
oSeq2	5′ GCAGGTCCTGAAGTTAACTAG 3′

Each plasmid was sent for sequencing with the oSeq1 and oSeq2 primers for Sanger sequencing (Eurofin Genomics, France).

## Data Availability

Mass Spectrometry data—LC-HRMS data collected from the bacterial cultures can be found on MASSIVE repository under the following identifiers: MSV000090468 (ftp://MSV000090468@massive.ucsd.edu, accessed on 5 October 2022).

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
