# Peer review of "Bioengineering of Escherichia coli Nissle 1917 for Production and Excretion of Spermidine, a Key Metabolite in Human Health"

_metabolites, 2022, doi:10.3390/metabo12111061_

Round 1
Reviewer 1 Report
I appreciate the study undertaken by Caffaratti et al. entitled “Bioengineering of Escherichia coli Nissle 1917 for overproduction and excretion of spermidine, a key metabolite in human health”. The study has significant implications and should be important in health sectors. I would request the authors to make clarifications regarding the following concerns that I have for this study:
1. The authors have used peak area to give an indication of the increased levels of production for spermidine, putrescine, etc. I would request the authors to cite other relevant literature that uses a similar approach.
2. Are there any specific biochemical assays that could be performed to show the change in production for the metabolites? If yes – the authors should carry out quick assays to support the production results. That would enhance the quality, standard, and reliability of the study.
3. The authors need to mention the magnitude of the overproduction of various metabolites in percentage for easy comprehension. I would ask authors to be more cautious and use the term “production” instead of “overproduction”.
4. Full names of the genes should be mentioned in table 1.
5. In the abstract – the authors should mention that this is the first evidence of spermidine production in a probiotic strain.
6. Line #57: Write the full name of EcN.
7. Line# 133: Bacterial samples or supernatant?
8. In statistical analysis (section: 2.7): include the number of replicates.
9. Are there any data for the authors' claim, “metabolite annotation” (subsection 2.6)? Please provide that.
Author Response
I appreciate the study undertaken by Caffaratti et al. entitled “Bioengineering of Escherichia coli Nissle 1917 for overproduction and excretion of spermidine, a key metabolite in human health”. The study has significant implications and should be important in health sectors. I would request the authors to make clarifications regarding the following concerns that I have for this study:
- The authors have used peak area to give an indication of the increased levels of production for spermidine, putrescine, etc. I would request the authors to cite other relevant literature that uses a similar approach.
We would like to thank you for your comment, we have cited relevant literature papers that uses a similar approach line 240 page 6 like this “(see the Methods section and reference papers [12,31]).
- Are there any specific biochemical assays that could be performed to show the change in production for the metabolites? If yes – the authors should carry out quick assays to support the production results. That would enhance the quality, standard, and reliability of the study.
That could be a good idea to determine the level of polyamines but to our knowledge, to date there is no rapid specific biochemical assay available to evaluate the level of polyamines. For this reason, we used an LC-MS/MS strategy that is a state-of-the-art methodology. In addition, our experiments were conducted using quality control samples, standards and sufficient independent replicates. Each replicate represents an independent culture. The experiments shown in Figures 2 and 3 compared to those shown in Figure 4 were conducted six months apart and we can see that the results are very robust and reproducible.
- The authors need to mention the magnitude of the overproduction of various metabolites in percentage for easy comprehension. I would ask authors to be more cautious and use the term “production” instead of “overproduction”.
Regarding the magnitude of the overproduction of various metabolites in percentage we added this in the section Result 3.3 line 320 and 321 page 10.
“Metabolic profiling of the different strains showed that the addition of the MdtI/MdtJ biological system induced a major change in the metabolite levels of the PAs pathway. Indeed, spermidine is significantly increased in EcN pCC31 and EcN pCC35 compared with EcN (Fold change: 3 and 5, respectively) or EcN transformed with pCC25 or pCC26 (Fold change: 2.8 between EcN pCC35 and EcN pCC25) (Fig 4D). Furthermore, the accumulation of acetylated forms of spermidine observed with plasmids pCC25 and pCC26 disappeared in strains transformed with plasmids pCC31 and pCC35 (Fig 4EF). These observations suggest that these engineering strategies profoundly impacted the metabolites fluxes belonging to the PAs family, in favor of spermidine production and excretion.”
We understand the remark on the term “overproduction” and we modified each time it was necessary :
Title Page 1 “Bioengineering of Escherichia coli Nissle 1917 for overproduction and excretion of spermidine, a key metabolite in human health.” was replaced by “Bioengineering of Escherichia coli Nissle 1917 for production and excretion of spermidine, a key metabolite in human health.”
Line 93 page 2 “Finally, we characterized the effect of the overproduction of spermidine in vitro” was replaced by “Finally, we characterized the impact of our engineering on spermidine production”.
Line 264 page 8, we modified the title of the figure 2 like this “Figure 2. Production of Spermidine by EcN pCC25 and EcN pCC26 compared to the wild type EcN. A Growth curves of EcN, EcN pCC25 and EcN pCC26 in LB medium (n=8)”
- Full names of the genes should be mentioned in table 1.
In the legend of the table 1, we added the following sentences in order to clarify the name of the proteins encoded by the genes, line 122 page 3.
“Table 1 : primers used in this study. speE (spermidine synthase) and speD (S-adenosylmethionine synthase). mdtJ and mdtI genes code for the spermidine export proteins MdtIJ. “
- In the abstract – the authors should mention that this is the first evidence of spermidine production in a probiotic strain.
As asked by the other reviewer, we revised entirely our abstract as following and add this sentence in particular : “Our results demonstrate, for the first time, the feasibility of rationally engineering bacterial probiotic strains to increase their ability to deliver the Microbiota-derived metabolite spermidine.”
- Line #57: Write the full name of EcN.
We have done this modification line 60 page 2.
- Line# 133: Bacterial samples or supernatant?
We have done this modification line 141 page 4.
- In statistical analysis (section: 2.7): include the number of replicates.
We agree that it was missing. We added this sentence “The individual data were obtained from 8 independent cultures.” Line 210 page 6.
- Are there any data for the authors' claim, “metabolite annotation” (subsection 2.6)? Please provide that.
As we understand the metabolite annotation part was not sufficiently clear, we added more details in the section 2.6 line 193 to 199 page 5, like that:
“The Library Search feature of GNPS allowed to identify N8-acetylspermidine (GNPS Library Spectrum CCMSLIB00006122363). In a second attempt to perform metabolite identification, SIRIUS and CSI :FingerID [25] were used. SIRIUS identifies metabolites based on their MS/MS spectra and by creating fragmentation trees that can explain the fragmentation spectrum. Then, CSI:FingerID uses the fragmentation tree to predict the compound’s molecular fingerprint. Using this second approach, we obtained a similarity score of, 100 for N8-acetylspermidine and, 92.1 for diacetylspermidine (See Supplementary Fig. 1&2).”
And we added two Supplementary Figures (Supplementary Fig 1 and 2) to illustrate our metabolite annotation strategy and results. Those results derived from the SIRIUS analysis.
Hence, we changed the numeration of our Supplementary Figures in the Section Result : 3.1, line 238 page 6: “We then used RT- 221 qPCR to relatively quantify the genes expression of speD and speE and we did observed a surexpression of speE and speD genes (Supplementary Fig 3).

Reviewer 2 Report
The authors investigated Escherichia coli Nissle 1917 (EcN) overproducing spermidine which is a kind of polyamines maintaining human health. They described that overexpression of speD and speE resulted in increase in the amount of spermidine. In addition, excretion of spermidine increased by increasing the expression of MdtI/MdtJ which is an exporter system of spermidine. The data was clearly presented, and the results will be of interest to people in the field. I think this is an interesting and important paper. However, a few points as indicated below need to be addressed by authors.
The abstract was vague and failed to explain the important strategic points of their study. They should describe it more concretely.
The section “6. Patients” (at the line 385) may be “5. Patients”.

Author Response
First of all, we would like to thank the reviewers for the time and effort spent on our manuscript. We appreciate the constructive and valuable comments that have been made. We believe that following the reviewers’ suggestions has substantially improved the quality of our manuscript. Please find our answer point by point :
The authors investigated Escherichia coli Nissle 1917 (EcN) overproducing spermidine which is one of polyamines molecules maintaining human health. They described that overexpression of speD and speE resulted in increase in the amount of spermidine. In addition, excretion of spermidine increased by increasing the expression of MdtI/MdtJ which is an exporter system of spermidine. The data was clearly presented, and the results will be of interest to people in the field. I think this is an interesting and important paper. However, a few points as indicated below need to be addressed by authors.
- The abstract was vague and failed to explain the important strategic points of their study. They should describe it more concretely.
We agree with the reviewer that this point requires clarification and modification. We proposed the revised version of our abstract.
“Microbiota-derived metabolites have biological importance for their host. Spermidine, a metabolite described for its protective effect in age-related diseases, is now studied for its role in the resolution of inflammation and gut homeostasis. Strategies to modulate its production in the gastrointestinal tract are of interest to increase host spermidine intakes. Here we show that metabolic engineering can be used to increase spermidine production by the probiotic Escherichia coli Nissle 1917 (EcN), used in humans. First, we found that increasing the expression of genes involved in polyamine biosynthesis, namely the S-adenosylmethionine synthase speD and the spermidine synthase speE, resulted in an increase of spermidine produced and excreted by our engineered bacteria. The major drawback of this first attempt was the production of acetylated forms of spermidine. Next, we propose to solve this problem by increasing the expression of the spermidine exporter system MdtI/MdtJ. This second strategy had a major impact on the spermidine profile found in the culture supernatant. Our results demonstrate, for the first time, the feasibility of rationally engineering bacterial probiotic strains to increase their ability to deliver the Microbiota-derived metabolite spermidine. This work illustrates how metabolomic and synthetic biology can be used to design and improve engineered Live Biotherapeutic Products, that have the ambition to be used in personalized medicine.”
- The section “6. Patents” (at the line 385) may be “5. Patents”.
We agree and changed line 406 page 12.

Round 2
Reviewer 1 Report
The authors have improved the manuscript.